# A Local Model and Experimental Verification of the Crossflow Filtration of a Polydispersed Slurry

**Qianyou Wang, Guolu Yang * and Jing Lu**

Sewage and sludge research center, School of Water Resources and Hydropower Engineering, Wuhan University, Wuhan 430072, China; qianyouwang2019@gmail.com (Q.W.); jinglu20180714@gmail.com (J.L.)
* Correspondence: guoluyang123@gmail.com; Tel.: +86-027-68772215

**Abstract:** In this paper, we propose a calculation model for a crossflow filtration process that is applicable to polydispersed slurry microfiltration. The deposition velocity of particles in slurry, particle distribution, and resistance of the filter cake on the surface of the filtration media can be predicted by this model, and can be used to predict the variations of filtration velocity. The theoretical prediction matched well with the experimental data, having a difference within 20%, except for the initial few seconds. However, the porosity of the filter cake used in the theoretical prediction was assigned based on the literature. It is revealed by the model that the variations in the crossflow filtration velocity are induced by the gradual domination of particles with small diameters in the filter cake. Meanwhile, the possible direction for the optimization of this model is pointed out.

**Keywords:** crossflow filtration; polydispersed slurry; filtration velocity; filter cake

## 1. Introduction

A special form of filtration that is widely applied in fields such as medicine, blood separation, sewage treatment, water purification, food, biotechnology, and petroleum [1–14] is called crossflow filtration. Crossflow filtration also exhibits promising application prospects in the separation of water and solid particles from sludge.

In this process, the filtration velocity gradually reduces to equilibrium, which is a major disadvantage of crossflow filtration. During the filtration process, the filtration velocity variation during the filtration must be accurately predicted in order to expand the industrial application of crossflow filtration.

Many theories have been proposed for predicting the filtration process in terms of uniform particle filtration. Trettin [15] introduced concentration polarization theory based on ultrafiltration and reverse osmosis. However, for micrometer-sized particles, the equilibrium filtration velocity based on the concentration polarization theory is generally one to two orders of magnitude lower than the experimental values [16]. According to this disagreement, it is inappropriate to consider Brownian diffusion as the only mechanism for the reverse diffusion of solid particles in the slurry during microfiltration. Considering the interaction forces between particles, Zydney [16] applied shear-induced diffusion in place of Brownian diffusion in concentration polarization theory to predict the filtration process of red blood cells and found that the prediction results were in good agreement with the experimental values. In addition, assuming a balance between inertial lift and hydrodynamic force in equilibrium, some researchers [17–19] proposed models that apply inertial lift instead of particle diffusion. According to experimental verification, the inertial lift theory is generally applied to solid particles larger than 30 μm [20]. Based on the hypothesis that the filtration velocities calculated from the inertial lift theory and concentration polarization theory are algebraically summable, Gutman [21]

carried out a study that theoretically proved that the results were accurate for slurry with a comparatively low solid volume fraction as applied by Altena and Belfort [22].

All of the aforementioned theories assume a narrow size distribution of solid particles. Indeed, slurries that are used for crossflow filtration normally contain particles with a wide distribution range. For instance, the solid particles in natural sludge range from the sub-micrometer to millimeter-scale in size. Therefore, it is of great importance to investigate crossflow filtration for polydispersed slurries.

With regard to the crossflow filtration of a polydispersed slurry, Dharmappa [23] considered the nonuniformity of slurry solid particles and applied inertial lift, shear-induced diffusion, and Brownian diffusion in the reverse movement mechanism of solid particles. However, an empirical formula was applied to determine the variation of the filter cake thickness over time, which was considered to be a semiempirical model that requires experimental results to determine the parameters needed to predict the filtration process. In addition, in their study, it was assumed that the particle size distribution was uniform in the filter cake along the thickness direction. Therefore, the average particle size was used to express the properties of the filter cake, such as specific resistance. This may result in errors in the prediction as confirmed in previous studies showing that the filter cake formed during crossflow filtration is immobile [10], which indicates that the solid particles that accumulate on the surface of the filter cake cannot be uniformly mixed with the layers deposited earlier.

Based on the results of Dharmappa, the theory of particle stability instead of the reverse transport theory was applied to the prediction by Foley [24]. In addition, the filter cake was considered to be immobile and layered, and this cake represents a comprehensive representation of the properties of each layer. However, the theory of particle stability only includes the hydrodynamic force on solid particles deposited on the filter media surface and excludes other forces between particles and also between particles and the filter media. Obviously, these forces are not negligible because the solid particles actually lie on the surface of the filter media. Meanwhile, only the friction coefficient was assumed to indirectly express the forces, and the model was not verified by experimental results.

Based on the results of Dharmappa [23] and Foley [24], inertial lift, shear-induced diffusion, and Brownian diffusion were all applied to the mechanism of the reverse movement of solid particles in this study. Moreover, the layering characteristics of the filter cake were also considered in order to predict the crossflow filtration process of polydispersed slurry.

## 2. Materials and Methods

### 2.1. Theory

The accumulation and resuspension of solid particles on the filter media's surface are used to control the thickness of the filter cake. The reverse transport process consists of three components: Brownian diffusion, shear-induced diffusion, and inertial lift.

Figure 1 is a schematic diagram of the crossflow filtration of the tubular membrane. The slurry inflows from the left side and outflows from the right side. The filtrate is generated under the pressure difference between the inside and outside of the tubular membrane. By assuming that the cake is incompressible with a constant void fraction, particles will be deposited on the surface of filter cake or filter media only if the reverse velocity is smaller than the filtration velocity. No particles are present in the permeate and the solid volume fraction in the slurry is much lower than that in the filter cake. The rate of cake formation on the membrane surface along the radial direction can be obtained while taking into consideration the mass balance during the accumulation process [24,25]:

$$\frac{\mathrm{d}\delta}{\mathrm{d}t} = \frac{\phi_0}{\phi_c} \sum_j \left(J - v_{rj}\right)_+ p_j, \tag{1}$$

where $\delta$ means filter cake thickness in m; $t$ means time in second; $\phi_0$ and $\phi_c$ mean the solid volume fraction in the slurry and in the filter cake; $J$ means filtration velocity in m/s; $j$ is a natural number and $p_j$ means the solid particle volume fraction of group $j$ in slurry; $v_r$ means the reverse velocity of the

solid particles in m/s; and $\left(J - v_{rj}\right)_+ = J - v_{rj}$ when $J > v_{rj}$ and $\left(J - v_{rj}\right)_+ = 0$ when $J \leq v_{rj}$. An explicit discretization of Equation (1) leads to

$$\Delta\delta(t) = \delta(t) - \delta(t - \Delta t) = \Delta t \cdot \frac{\phi_0}{\phi_c} \sum_j \left(J(t - \Delta t) - v_{rj}(t - \Delta t)\right)_+ p_j(t - \Delta t), \tag{2}$$

where $\Delta\delta$ means the increase of cake thickness per time step in m and $\Delta t$ is the time step.

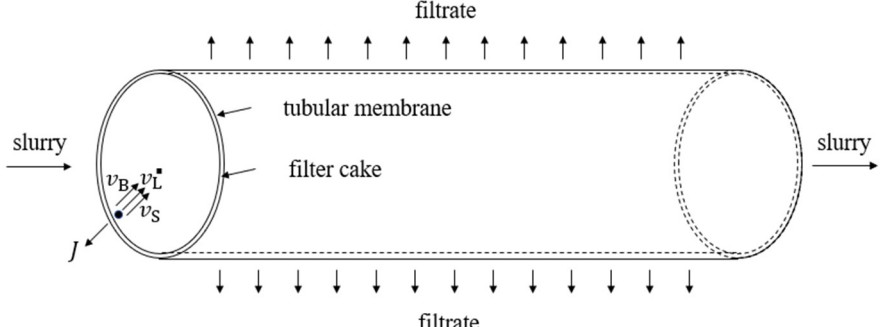

**Figure 1.** Schematic diagram of the crossflow filtration of the tubular membrane.

Assuming that the total solid volume fraction in the filter cake is constant and does not change along the thickness direction, according to Darcy's law [26,27], we obtain

$$J = \frac{\Delta P}{\mu(R_m + K_h\delta)}, \tag{3}$$

where $\Delta P$ means transmembrane pressure in Pa; $\mu$ means water viscosity in Pa·s; $R_m$ means resistance of the filter medium in 1/m; and $K_h$ means the filter cake specific resistance in 1/m$^2$.

The diffusion coefficient of Brownian diffusion can be expressed as [28]

$$D_{Bj} = \frac{K_b T}{3\pi\mu d_j}, \tag{4}$$

where $D_B$ means the Brownian diffusion coefficient in m$^2$/s; $K_b$ is the Boltzmann constant in J/K; $T$ means temperature in $K$; and $d_j$ indicates the solid particle diameter of group $j$ in m.

Moreover, the corresponding characteristic velocity can be expressed as,

$$v_{Bj} = \frac{D_{Bj}}{d_j} = \frac{K_b T}{3\pi\mu d_j^2}, \tag{5}$$

where $v_B$ refers to Brownian diffusion characteristic velocity in m/s.

Eckstein [29] conducted a comprehensive study on shear-induced diffusion by tracking the motion of individual solid particle. It was proposed that the shear-induced diffusion coefficient of solid particles varied linearly with the shear rate and the square of the particle radius. The diffusion coefficient is independent of the solid volume fraction when the solid particle volume fraction is greater than 0.2:

$$D_{Sj} = \frac{0.03}{4} d_j^2 \gamma. \tag{6}$$

The shear-induced diffusion characteristic velocity can be expressed as

$$v_{Sj} = \frac{D_{Sj}}{d_j} = \frac{0.03}{4} d_j \gamma, \tag{7}$$

where $D_S$ is the shear-induced diffusion coefficient in $m^2/s$; $v_S$ is the shear-induced diffusion characteristic velocity in m/s; and $\gamma$ is the shear rate in 1/s.

The inertial lift velocity of the solid particles in a laminar flow state can be expressed as [30]

$$v_{Lj} = \frac{b\rho d_j^3 \gamma^2}{128\mu},$$

(8)

where $v_L$ is the inertial lift characteristic velocity in m/s; $b$ is a coefficient; and $\rho$ means water density in $kg/m^3$.

In areas close to the filter media, a value of 0.577 [31] can be assigned to $b$ in a fast laminar flow; therefore, we obtain

$$v_{Lj} = \frac{0.0045\rho d_j^3 \gamma^2}{\mu}.$$

(9)

The reverse velocity of solid particles can be expressed as

$$v_{rj} = v_{Bj} + v_{Sj} + v_{Lj}.$$

(10)

During crossflow filtration, the area close to the filter media surface is considered for calculating the solid particles' reverse movement. When a filter cake is formed, the water's superficial velocity in the filter cake is estimated to be around $10^{-5}$ m/s [32,33], whereas the slurry velocity is estimated to be around $10^0$ m/s. The filter cake can be considered as part of the filter media because the liquid velocity in the filter cake is much lower as compared to that of the slurry. The turbulent viscous sublayer covers the filter cake's surface [34–36]. Therefore, the influence of turbulence is not considered when calculating the solid particle's reverse movement.

In summary, the particles of each size group correspond to a reverse velocity with the constant conditions of slurry temperature, flow rate, concentration, pressure, etc. The corresponding particles will accumulate on the filter media surface if the filtration velocity is higher than the reverse velocity or they will not be deposited on the filter media or filter cake.

To form a filter cake, solid particles with lower reverse velocity than the filtration velocity will accumulate on the filter media's surface. The layer thickness per time step is denoted by $\Delta\delta_i$, where $i$ represents the counts of the time steps, $i\cdot\Delta t = t$. The solid particles within the layer of $\Delta\delta_i$ can be considered as uniformly mixed when the time step is sufficiently small. The filtration resistance increases with thickening of the filter cake, which then reduces the filtration velocity to $J_i$.

According to Equation (3), to calculate the filtration velocity ($J$), it is necessary to obtain the specific resistance ($K_h$) of the filter cake. The filtration resistance cannot be calculated based on the average particle size of the solid particles because the size of the solid particles in the filter cake vary significantly, and the particle volume ratio corresponding to each size is also different. In the study of Kozeny–Carman [37,38], the total surface area of the solid particles that form the filter cake was converted into the surface area of circular pipes via Poiseuille flow. Then, the particle size was used to express the total surface area of the particles in the uniform filter cake. The specific surface area can still be applied to calculate the filter cake filtration resistance although the current study focuses on nonuniform solid particles. The corresponding equation is expressed as follows:

$$K_h = \frac{c_K(1-\varepsilon)^2 S^2}{\varepsilon^3},$$

(11)

where $c_K$ means the Kozeny constant; $\varepsilon$ means the filter cake porosity; and $S$ means the filter cake specific surface area in 1/m. The dimensionless Kozeny constant is calculated to be 5 and $\varepsilon = 1 - \phi_c$. In this way, the above equation can be converted into

$$K_h = \frac{5\phi_c{}^2 S^2}{(1 - \phi_c)^3}.$$ (12)

Assuming the solid particle has a sphericity of 1, the specific surface area is given by

$$S = \frac{\sum_j \left(J - v_{rj}\right)_+ p_j \Delta t \frac{6}{d_j}}{\sum_j \left(J - v_{rj}\right)_+ p_j \Delta t}.$$ (13)

In the above equation, $\left(J - v_{rj}\right)_+$ represents the velocity of the particles of group $j$ moving toward the filter media. The shear rate must first be obtained by the following equation in order to calculate the characteristic velocities of shear-induced diffusion and inertial lift:

$$\gamma = \tau / \mu,$$ (14)

where $\tau$ means shear force in N. The shear force in the equation can be approximated by the shear force at the filter media surface, i.e., $\tau = \tau_w$, where $\tau_w$ means wall shear force in N. Because the calculation of the particle reverse velocity only involves the area close to the filter media, the corresponding equation can be expressed as follows:

$$\gamma = \tau_w / \mu.$$ (15)

According to the friction velocity equation, we obtain

$$\tau_w = u_*^2 \rho,$$ (16)

where $u_*$ means friction velocity in m/s. The frictional head loss coefficient and the slurry velocity can also be used to express the friction velocity as follows [39]:

$$u_* = \sqrt{\frac{\lambda}{8}} v,$$ (17)

where $v$ means slurry velocity in the pipeline in m/s. In a circular pipe, the slurry velocity can be calculated as follows:

$$v = \frac{Q}{\pi r^2},$$ (18)

where $Q$ means the slurry flow rate in the pipeline in m$^3$/s and $r$ means the radius of the pipe in m. The pipe radius of the slurry flow varies as solid particles accumulate on the filter media's surface to form a filter cake, which can be expressed as

$$r = R - \delta,$$ (19)

where $R$ means the initial radius of the pipe in m. Since the slurry flow in this experiment is in a turbulent state, with the particle size being relatively small, the slurry flow can be determined according to the Colebrook's work for a turbulent smooth tube as for the frictional head loss coefficient [40]:

$$\frac{1}{\sqrt{\lambda}} = 2\lg\left(Re \sqrt{\lambda}\right) - 0.8,$$ (20)

where $\lambda$ is the frictional head loss coefficient and *Re* is the Reynolds number, which can be obtained as follows:

$$Re = \frac{\rho v 2r}{\mu}. \tag{21}$$

### 2.2. Solution Method

Initially, in order to determine the particle size distribution and the solid mass fraction, tests were carried out on the polydispersed slurry. The solid particles were divided into a limited number of groups based on the particle's sizes. In this paper, to emphasize the effect of particles with small diameter, natural logarithm is used to generate particle groups. An equidistant subdivision of the logarithmic values is used to identify the groups. The minimum particle diameter is set as 0.01 μm, the maximum is 200 μm, and the amount of the group is set as 50. Subsequently, the average particle size within each group and the solid volume proportion of each group in the total solid volume were determined.

Before we start each experiment, solid volume fraction in slurry and filter cake, transmembrane pressure, viscosity, temperature, Boltzmann constant, resistance of the filter medium, radius of tubular membrane, slurry flow rate, and water density are confirmed. An index *i* meaning *i*-th step is assigned to properties that will change along with time and $t = i \cdot \Delta t$. At $i = 0$, the filter cake was not yet formed from the solid particles and so both $\delta_0$ and $\Delta \delta_0$ equal 0 and the effective radius of cross section $r_0$ equals the inner radius of the tubular membrane *R*. The flow rate of slurry in pipeline (*Q*) and effective radius of cross section ($r_0$) are used to calculate the average velocity of slurry flowing in pipeline ($v_0$) according to Equation (18). Slurry density ($\rho$), viscosity ($\mu$), effective radius ($r_0$), and average velocity of slurry flowing in pipeline ($v_0$) are substituted to Equation (21) to calculate the Reynolds number ($Re_0$). The Reynolds number ($Re_0$) is substituted to Equation (20) to calculate frictional head loss coefficient ($\lambda_0$). Frictional head loss coefficient ($\lambda_0$) and average velocity of slurry flowing in pipeline ($v_0$) are substituted to Equation (17) to calculate friction velocity ($u_{*0}$). Friction velocity ($u_{*0}$) and slurry density ($\rho$) are substituted to Equation (16) to calculate shear force at the filter surface ($\tau_{w0}$). Shear force at the filter surface ($\tau_{w0}$) and slurry viscosity ($\mu$) are substituted to Equation (15) to calculate shear rate at the filter surface ($\gamma_0$).

Boltzmann constant ($K_b$), temperature (*T*), slurry viscosity ($\mu$), particle size distribution, and shear rate at filter surface ($\gamma_0$) are substituted to Equations (5), (7), (9), and (10) to calculate reverse velocity of solid particles in each group ($v_{rj,0}$). Transmembrane pressure, viscosity, resistance of the filter medium, average filter cake specific resistance per unit area ($K_{h0}$), and thickness of filter cake ($\delta_0$) are substituted to Equations (3) and (22) to calculate filtration velocity ($J_0$). Only if filtration velocity ($J_0$) is larger than reverse velocity of solid particles in one group ($v_{rj,0}$), solid particles will become part of filter cake. Reverse velocity of solid particles in each group ($v_{rj,0}$), filtration velocity ($J_0$), particle size distribution, and time step ($\Delta t$) are substituted to Equations (2) and (13) to calculate filter cake layer thickness ($\Delta \delta_1$) and average specific surface area of filter cake formed in this time step ($S_1$). Kozeny constant ($c_K$), solid volume fraction in filter cake ($\phi_c$) and average specific surface area of filter cake formed in this time step ($S_1$) are substituted to Equation (12) to calculate average filter cake specific resistance per unit area ($K_{h1}$). Cake layer thickness ($\Delta \delta_1$) and average filter cake specific resistance per unit area ($K_{h1}$) are substituted to Equations (22) and (3) to calculate filtration velocity ($J_1$). Filter cake thickness ($\delta_1$) and radius of tubular membrane are substituted to Equation (19) to calculate effective radius of cross section ($r_1$). This calculation moves to the next time step until the set time.

It is worth noting that while calculating filtration velocity ($J_i$) using Equation (3) at any time, the resistance of former filter cake should be included:

$$(K_h \delta)_i = \sum_i K_{hi} \Delta \delta_i, \tag{22}$$

Based on the results of Foley [24], the value of the filter's cake solid volume fraction was defined as 0.73.

*2.3. Experimental Section*

In Figure 2, the solid lines with arrows represent the slurry flow pipes, whereas the dotted lines with arrows represent the data transmission lines by which the data information, i.e., pressure, flow rate, and filtrate weight is transmitted to the computer for analysis. The slurry tank provides slurry for the pipeline and then recycles it from the pipeline. In order to maintain a consistent temperature of the slurry, tap water is sprayed on the outer surface of the slurry tank. The screw pump induces pressure and flow for the slurry in the pipeline, whereas the regulation of the crossflow filtration pressure and the flow rate is realized by sluice and ball valves. In order to obtain data on the flow rate, membrane pressure, and filtrate weight, respectively, the flow meter, pressure gauge, and balance are used. The screw pump was purchased from the local electromechanical market and has at most head of lift of 120 m and a flow rate of 0.0056 m$^3$/s. The pressure gauge has a measurement range of $6 \times 10^5$ Pa and a measuring precision of $1 \times 10^4$ Pa. The electronic balance was a Sartorius BSA124S with a range of 120 g and a measuring precision of 0.0001 g. This electronic balance can generate the filtrate weight every 0.95 s. A beaker with measuring range of 50 mL is placed on the electronic balance and is used to collect filtrate.

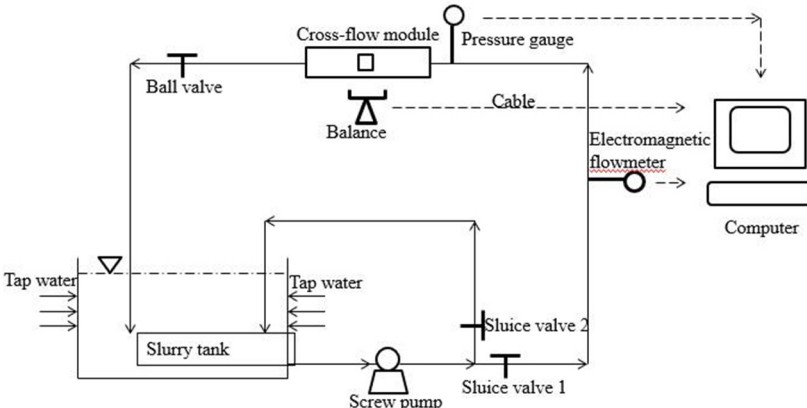

**Figure 2.** Schematic diagram of crossflow filtration experimental model.

Figure 3 is a photo of crossflow filtration module which consists of a polyvinyl chloride pipe with holes drilled on the wall and a filter cloth attached inside. Figure 4 features a schematic diagram of the supporting layer of the crossflow filtration module. In Figure 3, the bulges at both sides of the module are the fastenings used to fasten the module and the slurry pipe. A filter cloth is also attached to the walls of all pipes 1.5 m upstream and downstream from the crossflow filtration module to avoid an abrupt change of the flow state in the crossflow filtration area. The mesh number of the filter cloth is 2000 and the thickness is 0.001 m. In Figure 4, the length of the supporting layer is 0.1 m and the outer radius and inner radius are 0.016 m and 0.014 m, respectively. Holes with diameters of 0.003 m were drilled by a hand drill and six holes are distributed evenly at each circle along the peripheral direction. Along the axial direction, the distance between the edge and the center of the marginal drilled hole is 0.01 m, and the distance between the centers of two adjacent drilled holes is 0.016 m.

A Malvern Mastersizer 3000 manufactured by Malvern Instruments was used to measure the particle size distribution of the sediment samples, sludge slurry, and filter cake.

At the beginning of each experiment, a module that was exactly the same as crossflow filtration module but without drilled holes on the wall of the supporting layer was used. By adjusting the valves, the expected pressure and flow rate of the slurry were achieved. We kept the open angle of these valves unchanged and the power was then shut down, and this module was replaced by the crossflow filtration module. Then the power was reopened. All the pressure, flow rate, and filtrate weights were recorded. The slurry sample was taken at the outer mouth of the slurry pipe to measure the particle

size distribution by a Malvern Mastersizer 3000. Every time the beaker used to collect filtrate is about to be full, filtrate is poured to the slurry tank.

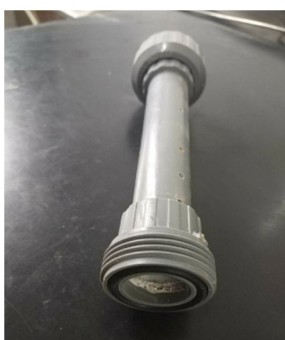

**Figure 3.** Photo of the crossflow filtration module.

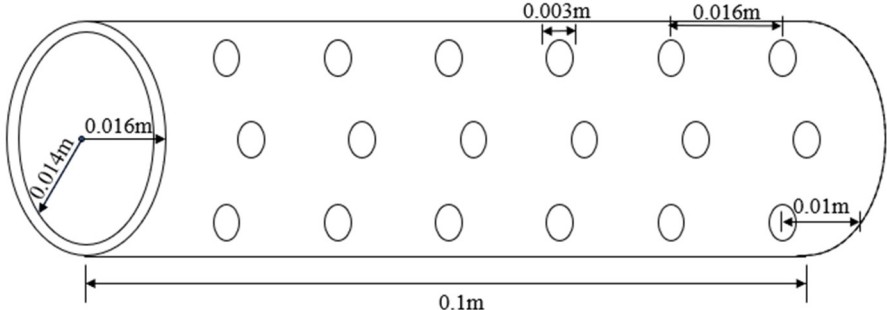

**Figure 4.** Schematic diagram of the supporting layer of the crossflow filtration module.

At the end of each experiment, the crossflow filtration module was immerged into water in a beaker. This beaker was weighed aforehand. A brush was used to move the filter cake into the water to form a slurry. This slurry was dried with a drying machine to obtain the weight data and the sample of this slurry was taken to measure the particle size distribution. Water density is considered to transform weight data of filtrate to volume data. Forward difference is used to obtain filtration velocity.

## 3. Results and Discussion

Figure 5 and Table 1 show the particle size distribution of the sediment sample. In Table 1, adjacent two numbers in row "size (μm)" form the upper and lower boundary of particle group. Figure 6 shows the cumulative curve of particle size distribution of the sediment sample. The particle size was estimated to range from 0.1 to 200 μm, as illustrated by a bimodal distribution. The average particle size is 1.13 μm and volume's average particle size was determined to be 5.21 μm, and the volume fraction of particles between 0.1 and 40 μm was more than 95%.

**Table 1.** Particle size distribution of the sediment sample in groups.

| size (μm) | 0.010 | 0.012 | 0.015 | 0.018 | 0.022 | 0.027 | 0.033 | 0.040 | 0.049 | 0.059 | 0.072 |
|---|---|---|---|---|---|---|---|---|---|---|---|
| $p_j$(%) | 0.00 | 0.00 | 0.00 | 0.00 | 0.00 | 0.00 | 0.00 | 0.00 | 0.00 | 0.00 | 0.00 |
| size (μm) | 0.088 | 0.108 | 0.131 | 0.160 | 0.195 | 0.238 | 0.290 | 0.353 | 0.431 | 0.525 | 0.640 |
| $p_j$(%) | 0.00 | 0.17 | 0.53 | 1.02 | 1.68 | 2.45 | 3.17 | 3.97 | 4.36 | 4.55 | 4.35 |
| size (μm) | 0.781 | 0.952 | 1.160 | 1.414 | 1.724 | 2.102 | 2.562 | 3.123 | 3.807 | 4.641 | 5.658 |
| $p_j$(%) | 3.70 | 2.91 | 2.12 | 1.73 | 1.69 | 1.94 | 2.27 | 2.76 | 3.10 | 3.56 | 3.92 |
| size (μm) | 6.897 | 8.408 | 10.25 | 12.50 | 15.23 | 18.57 | 22.64 | 27.60 | 33.64 | 41.01 | 50.00 |
| $p_j$(%) | 4.26 | 4.44 | 4.90 | 4.81 | 4.92 | 4.47 | 4.17 | 3.43 | 2.79 | 2.04 | 1.42 |
| size (μm) | 60.94 | 74.29 | 90.56 | 110.40 | 134.58 | 164.06 | 200.00 | | | | |
| $p_j$(%) | 0.97 | 0.64 | 0.41 | 0.26 | 0.13 | 0.00 | | | | | |

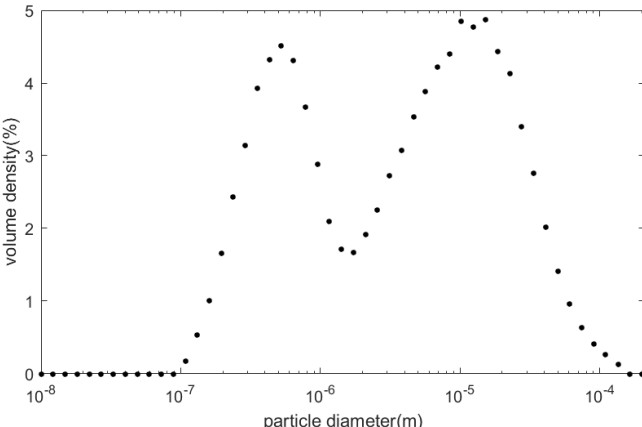

**Figure 5.** Particle size distribution of the sediment sample collected at the Yellow River Garden Mouth.

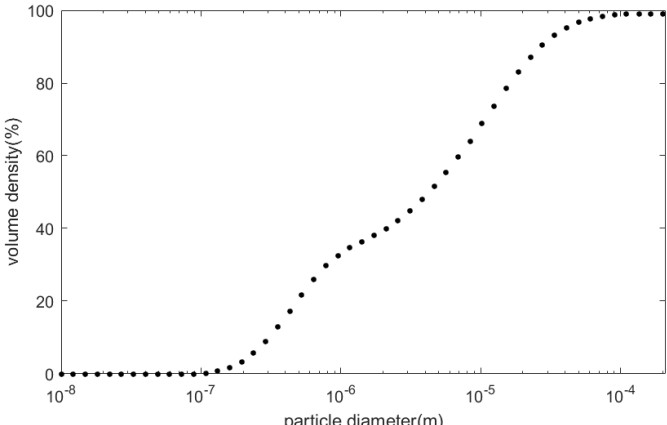

**Figure 6.** Cumulative curve of particle size distribution of the sediment sample collected at the Yellow River Garden Mouth.

Figure 7 shows the variation of the characteristic velocities of Brownian diffusion, shear-induced diffusion, and inertial lift with the sizes of the solid particles, which were calculated at a temperature of 27 °C and a shear rate of 12,000 s$^{-1}$. Within the size range of <0.2, >10, and 0.2–10 µm, Brownian diffusion, inertial lift, and shear-induced diffusion, respectively, were found to play major roles in the reverse movement of the particles.

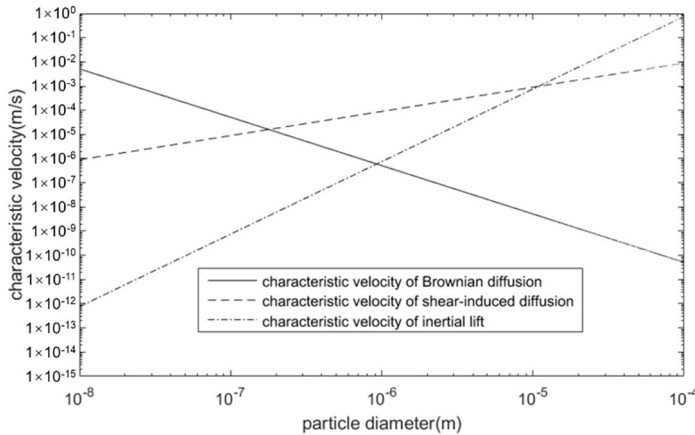

**Figure 7.** Variation of the characteristic velocities with the diameter of the solid particles.

The variation in the particle's reverse velocities with particle size, (calculated at a temperature of 27 °C and a shear rate of 12,000 s$^{-1}$) is displayed in Figure 8.

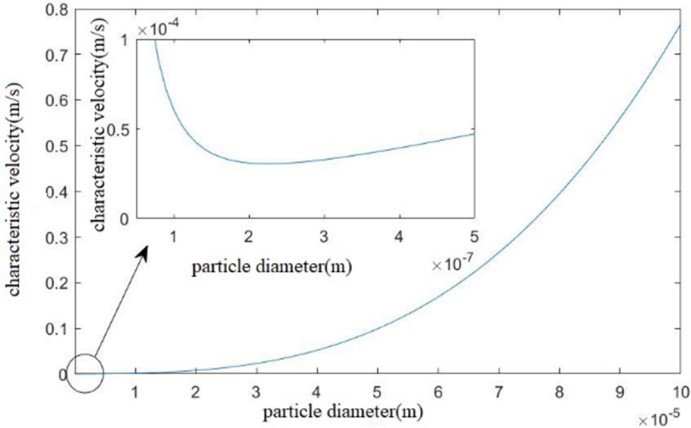

**Figure 8.** Variation of the reverse velocities with the diameters of the solid particles.

The minimum value of the solid particle's reverse velocity ($v_r$) is estimated when the effects of the characteristic velocities of Brownian diffusion, shear-induced diffusion, and inertial lift on the reverse movement of the solid particles are considered. Let

$$\frac{\mathrm{d}\,(v_r)}{\mathrm{d}\,(d)} = 0; \tag{23}$$

then, the particle size corresponding to the minimum reverse velocity can be obtained. At this time, particles with larger or smaller diameter than this value will not be deposited. If the thickness of the filter cake formed during the crossflow filtration is negligible compared to module scale, the minimum crossflow filtration velocity will also be achieved, which could be considered as the equilibrium state filtration velocity.

As shown in Figure 9, the experimental results and calculated results of the crossflow filtration velocity were compared. The experimental results were obtained at a viscosity of $9.32 \times 10^{-2}$ Pa·s, a slurry flow rate of $5.6 \times 10^{-4}$ m$^3$/s, a polydispersed solid particle volume fraction of 0.017, a slurry temperature of 27 °C, a pressure of 0.45 MPa, and a filter medium of a multifilament filter cloth with resistance of $1 \times 10^{10}$ m$^{-1}$. The calculated results were obtained at a time step of 0.1 s.

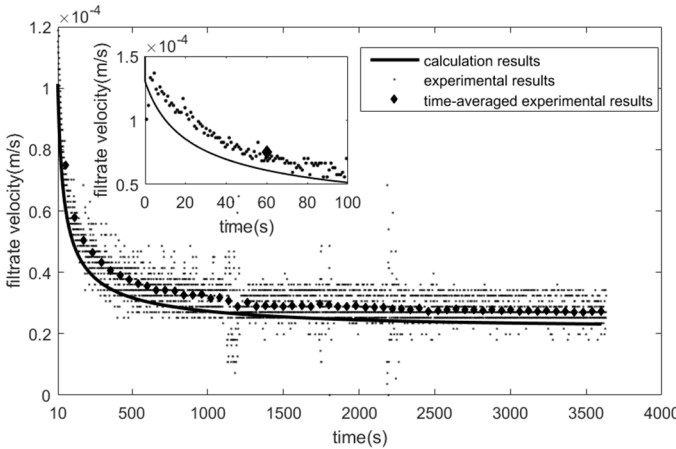

**Figure 9.** Comparison of the experimental results and calculated results of crossflow filtration.

The filtration velocity increases gradually in the first few seconds as shown by the experimental results. Because of the limitation of the experimental conditions, the target filtration pressure and slurry flow rate cannot be achieved instantaneously. Therefore, the filtration velocity increased gradually as the filtration pressure increased. After the initial stage, the experimental filtration velocity remained greater than the calculated result, with a moderately narrowed difference over time. At several typical time points (500, 1000, 2000, 3000, and 3600 s), the corresponding experimental velocity was $3.75 \times 10^{-5}$, $3.30 \times 10^{-5}$, $2.90 \times 10^{-5}$, $2.80 \times 10^{-5}$, and $2.70 \times 10^{-5}$ m/s, with the calculation results being $3.20 \times 10^{-5}$, $2.72 \times 10^{-5}$, $2.45 \times 10^{-5}$, $2.38 \times 10^{-5}$, and $2.32 \times 10^{-5}$ m/s, respectively. The associated discrepancy between the experimental and calculated values was found to be 14.7%, 17.6%, 15.5%, 15.0%, and 14.1%, respectively. Ultimately, the discrepancy was estimated to be within 20% during the test. The maximum difference between experimental filtration velocity and calculated filtration velocity in Dharmappa's work is 62%. As compared, the maximum difference in our work is 17.6%, which shows the advancement of our work. This discrepancy is acceptable given the fact that the current study minimized the constraints and reduced the assumptions of the study by Dharmappa and Foley. The calculated equilibrium filtration velocity was consistent with the experimental value when the experimental conditions were substituted into the equations.

As shown in Figure 10, the experimental particle size distribution results of the filter cake at 600 and 1200 s were compared during the crossflow filtration. The volume density of the solid particles with particle sizes smaller than 1 μm was found to be lower at 600 s than that at 1200 s. This phenomenon was caused by a decrease in the velocity that lowered the critical particle size below the size at which particles will deposited on the filter media. Therefore, the longer the crossflow filtration process, the higher the volume density of the small particles in the filter cake. For solid particles greater than 10 μm (particularly those greater than 100 μm) the particle size was found to be similar to that of the bubbles formed in water. Therefore, the volume density of large particles may vary significantly. In reality, this density does not reflect the composition of the solid particle in the filter cake. In addition, this result also confirms the conclusions proposed by Yoon [41], who determined that small particles play an important role in crossflow filtration equilibrium state.

As shown in Figure 11, the experimental and calculated results for the particle size distribution of solid particles in the slurry and the filter cake at 1200 s were compared. The result suggests a bimodal distribution of solid particles in the slurry in terms of their crossflow filtration with peaks at 0.5 and 15 μm, respectively. The value of the mean particle size was determined to be 4.73 μm. For solid particles in the filter cake, distribution with the peak centered at 0.5 μm was observed. The corresponding mean particle size was 1.14 μm, which was found to be smaller than the mean size of solid particles in the slurry. According to the experimental results, the volume density of the solid particles ranging from 0.01 to about 1.2 μm in the filter cake was greater than that in the slurry, whereas a lower volume density was observed for larger solid particles (1.2–35 μm) in the filter cake. This demonstrates that smaller solid particles are more likely to be deposited on the filter media's surface in the crossflow filtration process, thereby resulting in a relatively high fraction in the filter cake. Compared with particle size distribution in the slurry, the volume ratio of small particles (smaller than 1 μm) is larger and the volume ratio of large particles (larger than 1 μm) is smaller. However, the experimental and calculated particle size distribution of the filter cake are not consistent. One of the possible explanations is that the characteristic velocities of shear-induced diffusion and inertial lift are overrated. According to Zydney's work [16], the coefficient $\frac{0.03}{4}$ is true only when the volume fraction of particles exceeds 20%. As an immobile filter cake forms, the volume fraction at both side of filter cake surface is not continuous. The particles volume fraction in slurry closed to filter cake surface is less than 20%. As a consequence, the characteristic velocities of shear-induced diffusion is smaller than that we provided in this paper.

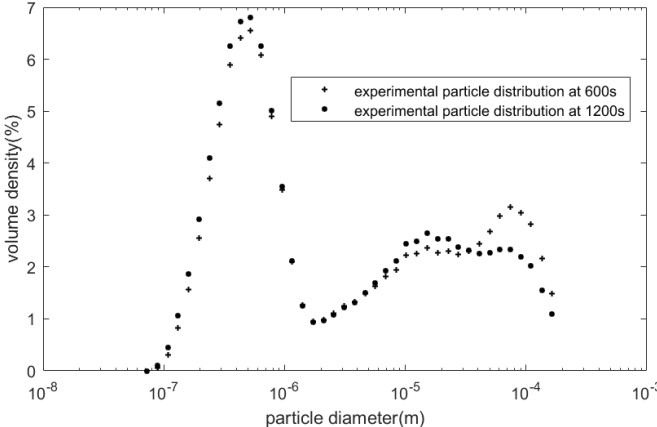

**Figure 10.** Particle size distribution of the filter cake at 600 and 1200 s.

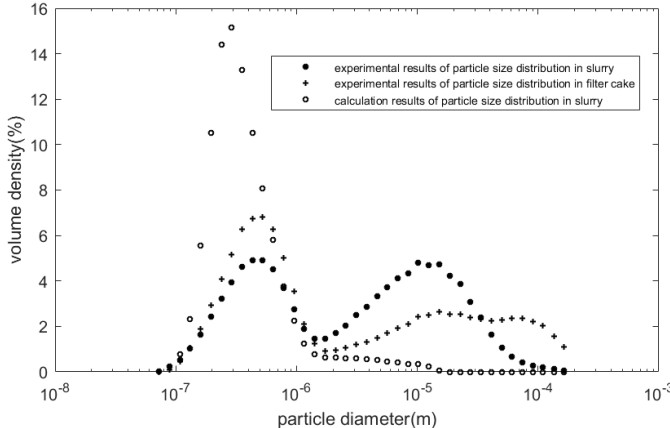

**Figure 11.** Size distribution of solid particles in the slurry and filter cake at 1200 s.

In Figure 12, the filter cake masses for the calculation and experiment are compared during the crossflow filtration process. The five triangles represent the measured filter cake mass based on experiments at 60, 180, 600, 1200, and 2400 s, respectively. As shown in the figure, the filter cake mass based on the calculation increased rapidly during the initial stage because of the relatively large filtration velocity, thereby leading to a distinct difference between the filtration velocity and the solid particle's reverse velocity; in this case, more particle size groups would accumulate on the filter media's surface. The difference between the filtration velocity and the solid particle's reverse velocity reduced as the filtration velocity decreased; consequently, fewer particle size groups were accumulated on the filter media surface to form the filter cake. As a result, the increased rate of the filter cake's mass reduced gradually. Eventually, solid particles stopped being deposited on the filter media's surface upon reaching equilibrium. Therefore, the filter cake's mass remained constant.

Comparing the calculation and experimental results of the filter cake mass, it was observed that the experimentally measured mass was greater than the calculated result at 60 s. This difference was caused by the limitations of the operating conditions. Solid particle deposition occurred even at very low pressure because the target filtration pressure and slurry flow rate cannot be obtained instantaneously in the initial stage, thereby resulting in a larger experimental value compared to the calculation result. Ultimately, the calculation results were estimated to be consistent with the experimental results at 180, 600, 1200, and 2400 s as the filtration process proceeded.

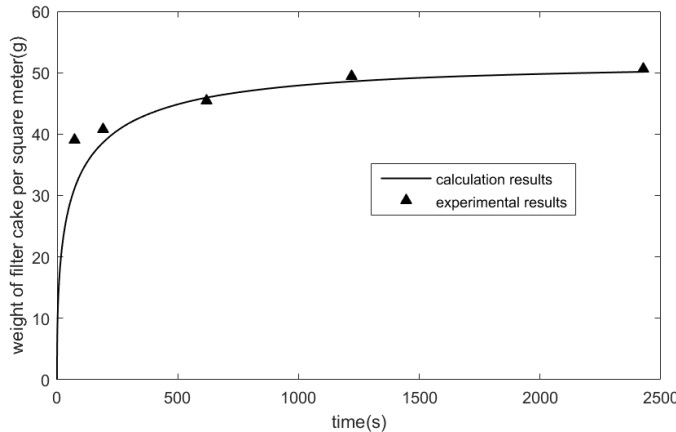

**Figure 12.** The predicted and experimental filter cake mass versus time.

## 4. Conclusions

Based on the results of previous studies, this paper considered the sum of the characteristic velocities of Brownian diffusion, shear-induced diffusion, and inertial lift to represent the reverse velocity of the solid particles that move from the filter media's surface to the slurry during the crossflow filtration process. By combining the Kozeny–Carman equation, the principle of mass conservation, and Darcy's law, an equation to determine the crossflow filtration of the polydispersed slurry was derived. The equilibrium of crossflow filtration was also proposed.

By comparing the calculated and experimental results for the velocity of crossflow filtration, the particle size distribution of the filter cake, and the filter cake's mass, the present model demonstrated high accuracy. Meanwhile, this model also predicted the equilibrium of the crossflow filtration of the polydispersed slurry, demonstrating good consistency with the experimental results. Importantly, the results of Yoon [41] that small particles play an important role in crossflow filtration equilibrium state were also validated by comparing the calculated results derived from the model with the experimental results.

Compared with the results of the previous studies, the present model entails fewer assumptions and can be applied with fewer limitations despite the limitations of the operating conditions. This study can be further improved in a few ways. For instance, the porosity of the filter cake was assigned by a constant value, and the internal pollution of the filter cloth was not considered. A more accurate prediction of the crossflow filtration process also depends on further research on the particle group movement in the slurry.

**Author Contributions:** Data curation, Q.W.; Formal analysis, Q.W.; Funding acquisition, G.Y.; Methodology, J.L.; Project administration, J.L.; Resources, J.L.; Software, Q.W.; Supervision, G.Y.; Validation, Q.W.; Writing—original draft, Q.W.; Writing—review and editing, J.L. All authors have read and agreed to the published version of the manuscript.

**Funding:** This research was funded by Wuhan University.

**Acknowledgments:** We are grateful for the financial support provided by Wuhan University.

**Conflicts of Interest:** The authors declare no conflict of interest.

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
