# Peer review of "A Local Model and Experimental Verification of the Crossflow Filtration of a Polydispersed Slurry"

_water, doi:10.3390/w12020489_

Round 1
Reviewer 1 Report
The authors provide a comparison between experimental results for polydisperse cake build up in cross-flow filtration and phenomenological formulas based on "the cake equation".
However, a lot of important information are missing (see in detail below) and therefore there is space for improvement. Unless those open points are clarified, the manuscript should not be published.
Major issues:
1) The list of references is quite spare. The article has 28 references. Fourteen of them are mentioned in the first sentence for showing the wide area of application of cross-flow filtration. In the following, several statements are not proved with references. As an example, the theory subsection (2.1) has only two (!!) references. However, most of the formulas are not original research work and therefore, references should be given. Those formulas were not only used in cross-flow filtration but also in dead-end filtration and thus also references from this field needs to be included.
2) There might be discrepancies between the theory section and the experimental setup. It seems that the theory sections is written with having a flat membrane in mind. However, the later application is a tubular membrane setup. This should be clarified. Maybe a sketch of the setup might help here.
3) The experimental setup is not explained. In subsection 2.3. a picture of the cross-flow filtration module is shown, however, no physical dimensions are given. Since the experimental setup is a major part of the contribution, those information need to be given. Without those dimensions a comparison to other experiments is not possible and the theory might not be applied correctly.
4) This continues with the exection of the experiments. In section 3 the results of the experiments are shown but it is not explained how the experiments are carried out. Also some of the experimental conditions are missing. As before: As the experiments are a major point of the contribution, more details need to be added. For example it is never explained how the particle size distribution of the filter cake is obtained. Please add the description of the experiments and their evaluation for all results and figures.
5) There are some notations and terminologies that are not used correctly. For example flow rate and velocity. This gives the impression that the authors have not worked carefully.
6) The mathematical computations in subsection 2.2. are only explained very shortly and the given information is not sound.
Further issues:
Line 14: What does "unqualified experimental conditions" mean? Since this might not be clarified in the abstract, it might not be mentioned there in this way.
Line 34: The authors are stating that flow rate differs between theory and experiments. This statement needs at least one reference.
Lines 61-63: Previous studies are mentioned but there are no references.
Line 70: mean while -> meanwhile
Lines 81-82: An assumption for the volume fractions of the slurry and the filter cake is made. Where is this assumption used and why the assumption is needed?
Line 82: Sentence beginning at the end of the line. What is it? The cake thickness?
Formula 1: What is summation index j? Number of particle sizes?
Line 84: The unit of flow rate is m^3/s and not m/s. This is velocity!!!
Line 85: Does p_j result from the particle volume distribution? Is there a difference of the distribution in cake and slurry (at least Figure 9 suggests this)?
Formula 3: I guess d should be d_j since we are considering a polydisperse slurry. Therefore also the following properties should all have an index j (corresponding "group j").
Line 92: K is doubled
Formula 6: Are the small d in formula 5 and 6 different? At least the style is different. Please clarify.
Formula 7: Why introduce a and not stay with d since a=d/2?
Line 105: Is the theory only limited to water since \rho is water density? I guess the theory holds for any fluid?! At least it should be consistent with \mu.
Lines 107-109: This statement might depend (from my point of view) on the setup. In the following a tubular cross-flow filtration module is used. Therefore the sign of b should be different at the top and the bottom of the tube (since also the filtration velocity has a different sign). This statement might only be true, if a cylindrical coordinate systems is used and the considerations are only made in radial direction. However, Figure 1 might indicate a flat setup.
Line 110: This is the reverse velocity of group j?!
Lines 113-114: This statements needs to be proved with references.
Lines 115-118: Is there any indication for a turbulent boundary layer? A lot of works (for example the works of Ripperger et al. which are not mentioned at all) do not consinder a turbulent boundary layer at all. If there is a turbulent boundary layer this also needs to be proved with references. Figure 1: What is the coordinate system, flow direction?
Lines 133-135: Considering this statement: Is it then correct to assume a constant porosity of the cake? Additionally: The authors are stating that the particle volume ratio is differing, how its value in formula (1) is chosen?
Lines 136-140: The Kozeny-Carman formula is very well known. However, I also suggest to give here some references.
Formula 10: What is i here? Do I need \epsilon_i?
Formula 12: What is summation index k? If I am correct, the unit of S_i should be 1/m^2. However, I do not get this unit from the formula. Since velocities are included, there is 1/s which is not cancelled. How the volume of the solid particles in the cake is calculated? If v_k is larger than J_i, then S_i might be negative. Is this possible?
Line 146: What does "level k" mean?
Formula 13: \delta_i is the thickness of layer i? Summation over the cake layers? How the layers are distinguished? Or is it summation over time steps? At which time step we are in the moment? Is \delta_t the current cake thickness?
Lines 150-153: It is stated that for the calculation of the reverse velocity only the area close to the filter media (=membrane) is considered. However, Fig. 1 indicates that the cake might be quite thick. Maybe this should be an assumption.
Formula 17: What is \lambda?
Line 157: Flow rate is always volumetric and has the unit m^3/s. v is velocity, Q is flow rate!
Lines 162-163: The authors need to prove that the flow is turbulent. Since no dimensions of the device are given, the reader can not compute a Reynolds number and therefore it is completely unclear why this is the case. If there is a reference for this statement, it should be given.
Formula 20: Why this formula is given? What is computed from this formula? \lambda? It is not used in the following. Should it be used in formula (17)? Is there any reference for this formula or is it original research of the authors?
Line 167-170: How the test are carried out? How the division is made?
Line 171: Should it \delta_0?
Line 174: What does v_j1 mean? Reverse velocity at time 1?
Line 176: Again. Please clarify flow rate (m^3/s) and velocity (m/s)!!!
Lines 177-178: It is stated that "the volume of the solid particles accumulated on the filter media surface per unit time step were calculated". How this is done?
Lines 178-185: From a mathematical point of view, it seems that the authors used explicit computations, since for updating to time step t+\Delta t only the values from time step t are used (something like this needs to result from the discretization of formula (1) ). How the time step \Delta t is chosen here?
Line 189: A value of "0.73" is stated. From my point of view this value should depend on the composition of the slurry and the particle size distribution of the cake.
Figure 2: The figure is too small. Not all details are visible.
Figure 3: More details! Length scale. I can not see the holes. What is the size of the holes?
Lines 209-212: Those measurements were done using the Malvern Mastersizer 3000 mentioned before? Since the measurement device also gives the mean particle size (by number) this value might be stated. Also cumulative curve helps to show that 95% are below 40 µm.
Lines 215-219: What fluid is used here? Water?
Figure 5: Here the scale is in m in Figure 4 it was in µm. Please stay with the same scales (also Figure 8).
Lines 228-232: This is correct for one particle size but not for all. An equilibrium state should be reached when all particle sizes satisfy formula (22).
Line 234: Stay with SI-units if stated so before. (m^3/h -> m^3/s)
Line 235: The solid volume fraction of only 1.7% seems to be very low. I doubt that this is slurry. Please clarify.
Figure 7: How many experiments were performed for those data? In the zoom the points indicate experimental results?
Lines 239-240: The authors state the filtration velocity (not the flow rate!!!) increases in the first few seconds. Do they mean the first three points in the zoom? Overall there is a decrease. Please clarify.
Lines 245-246: The scientific notation is not correctly. If E is used, no superscript is needed, e.g., E-5.
Lines 244-252: Is there a systematic error in the model? Since still some assumptions are made (as the constant porosity maybe), this might give directly an error of 10%.
Line 253: The authors state the particle size distribution was "calculated". For clarity: Was it measured (using the Mastersizer) or calculated from some formula?
Line 260: The authors state that there are bubbles in the water. Is there any indication for this during the experiments?
Lines 261-264: The authors state that the result is consistent with the model. Where is this shown? Which figure proves this statement? Please clarify.
Figure 8: How the particle size distribution in the cake is measured? Is the experiment stop and the particle size measuring device is used? Is it calculated based on some assumptions? Please clarify?
Line 271: From figure 9 I would not say that the particle size dsitribution in the filter cake is unimodal.
Lines 276-283: This seems not plausible for me. Please clarify.
Line 308: repersent -> represent???
Reviewer 2 Report
This manuscript reports a new model on cross flow filtration of poly-dispersed slurry suspension. It looks interesting, but it is not easy to catch up the original contribution of this model. At the end of introduction, the authors point out that previous studies did not consider some of the forces which are not negligible (p2, line 68), however this manuscript does not consider these forces too. The authors point out the limitations of previous models, but it is not clear how the authors really made a progress. It will be necessary to directly compare the models and show the predictions of both current model and previous models.
What is the criterion to define groups? It is too descriptive. It would be better to provide a diagram. In experiments, it is necessary to provide information on supplier and length scales of the apparatus. Why the sediment is taken from the yellow river garden mouth? Is it a standardized or well recognized sample? How much the test is reproducible? What is the operational condition? Why the data is provided at shear rate 12000s-1? In figure 8, it would be better to add the results of Dharmappa and Foley. What is the advantage of this model over previous ones? The size distribution seems to tri-modal. Is it possible to expand the model to tri-modal or more? If then what would be the difference? It is necessary to provide parameter values used in comparison with experiments.Author Response
Please see the attachment.

Round 2
Reviewer 1 Report
Thank you for providing a new version of your manuscript. It has improved tremendously, however, there are still a number of open points. Those should be clarified before publication. Additionally, a throughout formatting of the text is necessary.
Further questions arising from the issues of the first review, can be found in the attached document
Major issues:
1) The newly introduced formulas (21) and (22) might not be correct. First of all a sum from i=1 to i is not very meanigful.
I guess that the authors want to state in equation (21) that they want to add the resistances of the single layers of the cake. However, following equation (1) and the explanations in the lines 178-206, \delta_i is indicating the cake thickness at time step i. Therefore it is not the thickness of one of the layers. In detail: A discretization of equation (1) (explicit) leads to
\delta(t + \Delta t) = \delta (t) + \Delta t \cdot \\phi_0 /\phi_c \sum_k (J(t)-v_{rk}(t))p_k(t).
As it seems before an index stands for the next time step, therefore \delta_i = \delta(i\cdot \Delta t)?!
In lines 132-133 it is stated that \delta_i is the layer thickness at time t=i\cdot \Delta t. However, this is confusing for the reader. How the (total) cake thickness at time i is indicated and how the thickness of the i-th layer? The layer thickness might be given as \Delta \delta_i = \delta_i - \delta_{i-1}. For a better understanding, a difference should be made here.
This continues to equation (22).
2) Explain the notation that the index might be the actual time step (see your explanations in the lines 178-206 and assigning an additional index to all properties).
3) In all plots of the particle volume distribution it seems that you show the complete distribution. However, the complete theory is made for particle groups. How the discrete distributions look like?
4) The references at the end of the manuscript are not numbered, however, the referencing is made using numbers. Therefore, please use a consistent reference style.
5) Please unify the terminology of cross-flow filtration. In the text one can find the variants cross-flow, crossflow, and cross-filtration.
6) If the authors want to make a difference in the use of j and k for slurry and cake, respectively, they should also mention this in the text.
7) Maybe I have overlooked this, but it should be stated that the article assumes that the particles are spherical.
8) I appreciate that you want to give reference to the original research works of Darcy, Kozeny, and Blasius, however, I doubt that you looked into those works (in French and German) to get the formulas and information you used here. Please refer to any textbook or introduction to flow in porous media or turbulent flow (which you might have used). This is in any case more helpful than the references to the original works. Looking to \url{https://en.wikipedia.org/wiki/Darcy_friction_factor_formulae#Blasius_correlations}, I also guess that the reference to Blasius for equation (19) might not be correct. Please refer to the article or book you have used here.
9) The text still needs some formating. There is often a blank in front of the references missing. The font of "where" following an equation (e.g. equation (1), (2), ...) seems to be different.
Further issues:
Line 31: based on of --> based on
Lines 91-92: Please clarify use of i and k.
Line 100: If you state for all variables its unit, you should also state it for the Boltzmann constant K_b
Line 101: Should be d_j
Formula 6: d_j in denominator
Formula 9: Is the double index a misprint?
Line 141: The correct terminology is Poiseuille flow or better Hagen–Poiseuille equation
Formula 10 and 11: Here S seems to be used with small letter, however, capital letter might be appropriate.
Formula 10: The use of a small k for Kozeny constant might be confusing since k is used as summation index. Consider using capital K.
Lines 137-140, Line 148 and formula 12: You are stating that for a polydisperse sample with a broad distribution, an average diameter can not be used. However from my point of view also the specific surface is nothing else than an average diameter. You are defining a modified particle size distribution, which is given by (J-v_{rk})p_k \Delta t and then taking 6 divided by the weighted harmonic mean (\url{https://en.wikipedia.org/wiki/Harmonic_mean#Weighted_harmonic_mean}), compare also the works of Alderliesten, \textit{Mean particle diameters, Part I - Part VIII}, especially Part I and Part II.
For spherical particles one usually states that the specific surface is given as S_V = 6/D_{32}, where D_{32} is the Sauter mean diameter. If a kind of specific surface is intended, why not the weighted surface area and the weighted volume are related in a harmonic sense, i.e.,
S = \frac{6 \sum_{k} \frac{1}{d_k^3} (J-v_{rk}) p_k \Delta}{\sum_{k} \frac{1}{d_k^2} (J-v_{rk}) p_k \Delta}?
Please clarify terminology and usage.
Line 153: Font of symboles seems to be different.
Line 161: Q means the slurry flow rate
Line 181: effective radius r=r_0
Line 184: The authors are using slurry density. Is this different to water here? Slurry density might be understand in the sense of multiphase flow, where mixture density is computed using the arithmetic mean of liquid and solid density. For the multiphase viscosity mostly the harmonic mean is used. Please clarify if the data from water are used.
Lines 233-235: I do not understand why a filter cloth somewhere upstream and downstream (1.5m distance in comparison to 0.1m length of the module is quite far away) should help to influence the flow characteristics in the module. If there is already a filter cloth before, how do the authors adjust the results if particles might be already filtered here? Please clarify this setup.
Lines 271-278: I guess the velocities were calculated from the given formulas and not measured experimentally. If otherwise, please state how the measurements are made and how the different velocities can be distinguished in the experiments. The same question applies in line 277.
Figures 10 and Figure 11: Is the volume density really in percentage?
Figure 2: What does the y-label mean?
Lines 370-371: In which way the Kozeny-Carman equation is connected to mass conservation? From my point of view, it is a phenomenological formula for the resistance of a packed bed of spherical particles.
Line 377: If you give credit to the work of Yoon, please add the reference and also repeat the result that small particles have a large influence on the cake resistance.
Figure 2: A more general question. Since the outflow of the cross-flow module is going back into the slurry tank (kind of multi-pass test), is the concentration in the tank controlled? If the volume is not large enough, I would guess that the concentration is increasing.

Reviewer 2 Report
It is much improved and readable.
Round 3
Reviewer 1 Report
Thank you for the throughout revision of the manuscript. The text and the mathematical description is much clearer now. Using only one volume density makes it much clearer.
There are only a few minor issues left which should be resolved before publication:
Equation 13: Replace the top summation index J by j
Line 175: Add a blank: 200 µm
Line 194: If you refer to multiple equations please write "Equations"
Line 197: Same as before
Line 200: Here it should be cake layer thickness (since you are pointing to the increment). If you start with a zero thickness it holds that the increment is equal to the total cake thickness in the first step, however, to make it clear the right terminology should be used here.
Line 204: I would prefer "cake layer thickness" instead of "layer cake thickness"
Line 223: Add a blank: 0.0056 m^3/s
Line 225: Add a blank: 0.0001 g
Line 226: Add a blank: 0.95 s
Lines 230-240: Please add a blank when stating any dimensions, e.g. 1.5 m (compare how you did in lines 262-267 for the particle sizes).
Line 233-235: Did I get from your comments correctly that the filter cloth is attached at the wall of the pipe? If yes, than you might write "A filter cloth is attached to the walls of all pipes upstream and downstream from the crossflow filtration module."
Line 251: flowrate -> flow rate
Line 262: shows -> show
Line 299: Add a blank: 0.1 s
Since I may not have found all missing blanks in front of units, please check again the manuscript.
Reference 37: If you cite the original title of the work of Kozeny, it should be "Über" (https://en.wikipedia.org/wiki/%C3%9C)
